# Grapefruit Root and Rhizosphere Responses to Varying Planting Densities, Fertilizer Concentrations and Application Methods

**DOI:** 10.3390/plants12081659

**Published:** 2023-04-15

**Authors:** John M. Santiago, Davie M. Kadyampakeni, John-Paul Fox, Alan L. Wright, Sandra M. Guzmán, Rhuanito Soranz Ferrarezi, Lorenzo Rossi

**Affiliations:** 1Indian River Research and Education Center, Horticultural Sciences Department, Institute of Food and Agricultural Sciences, University of Florida, Fort Pierce, FL 34945, USA; 2Citrus Research and Education Center, Soil, Water and Ecosystem Sciences Department, Institute of Food and Agricultural Sciences, University of Florida, Lake Alfred, FL 33850, USA; 3Indian River Research and Education Center, Soil, Water and Ecosystem Sciences Department, Institute of Food and Agricultural Sciences, University of Florida, Fort Pierce, FL 34945, USA; 4Indian River Research and Education Center, Agricultural and Biological Engineering Department, Institute of Food and Agricultural Sciences, University of Florida, Fort Pierce, FL 34945, USA; 5Department of Horticulture, University of Georgia, Athens, GA 30602, USA

**Keywords:** citrus greening, *Citrus paradisi*, flatwoods, plant nutrition, rhizosphere

## Abstract

Huanglongbing (HLB) disease has caused a severe decline in citrus production globally over the past decade. There is a need for improved nutrient regimens to better manage the productivity of HLB-affected trees, as current guidelines are based on healthy trees. The aim of this study was to evaluate the effects of different fertilizer application methods and rates with different planting densities on HLB-affected citrus root and soil health. Plant material consisted of ‘Ray Ruby’ (*Citrus × paradisi*) grapefruit trees grafted on ‘Kuharske’ citrange (*Citrus × sinensis × Citrus trifoliata*). The study consisted of 4 foliar fertilizer treatments, which included 0×, 1.5×, 3× and 6× the University of Florida Institute of Food and Agriculture (UF/IFAS) recommended guidelines for B, Mn and Zn. Additionally, 2 ground-applied fertilizer treatments were used, specifically controlled-release fertilizer (CRF1): 12−3−14 + B, Fe, Mn and Zn micronutrients at 1× UF/IFAS recommendation, and (CRF2): 12−3−14 + 2× Mg + 3× B, Fe, Mn and Zn micronutrients, with micronutrients applied as sulfur-coated products. The planting densities implemented were low (300 trees ha^−1^), medium (440 trees ha^−1^) and high (975 trees ha^−1^). The CRF fertilizer resulted in greater soil nutrient concentrations through all of the time sampling points, with significant differences in soil Zn and Mn. Grapefruit treated with ground-applied CRF2 and 3× foliar fertilizers resulted in the greatest bacterial alpha and beta diversity in the rhizosphere. Significantly greater abundances of Rhizobiales and Vicinamibacterales were found in the grapefruit rhizosphere of trees treated with 0× UF/IFAS foliar fertilizer compared to higher doses of foliar fertilizers.

## 1. Introduction

Much citrus production worldwide has significantly declined due to a disease known as huanglongbing (HLB, or citrus greening) [1]. In Florida, citrus production has been reduced by more than 70% since HLB was first detected in 2005 [2,3,4]. The disease is associated with *Candidatus* Liberibacter asiaticus (*C*Las), bacteria transmitted to tree hosts by a vector called the Asian Citrus Psyllid (*Diaphorina citri*, ACP) [5,6,7]. As the bacteria colonize the sieve tubes within the phloem, callose deposition occurs, resulting in the plugging of the phloem, thus inhibiting nutrient uptake [1]. Symptoms following infection can include leaf chlorosis, leaf drop, reduced canopy density, smaller fruit size, root dieback, lack of juice quality and reduced yield [8]. Grapefruit, in particular, are more prone to root dieback when infected with HLB compared to specialty citrus, such as lemon and lime [9]. Most of the grapefruit production in Florida takes place in the Indian River district, a 200-mile-long area that borders the Atlantic Ocean. Since the introduction of HLB, grapefruit production has undergone an 85% decline [10].

There are several approaches toward dealing with HLB and alleviating symptoms, including the development of disease-tolerant rootstocks [7,11], the implementation of vector control [12] and the use of soil amendments [13] to improve soil health. The effect of different methods and rates of fertilizer application on citrus health and yield have also been studied [4,14,15,16]. There are several ways of applying nutrients to citrus: foliar applications, ground-applied granular fertilizers, fertigation and banding [14]. The mobility of nutrients in soil and plant tissues is a notable factor in deciding which application method is most appropriate [15]. For instance, when applying micronutrients (generally performed in smaller quantities), the efficiency of root uptake may be compromised when soil conditions are not favorable, and thus the foliar application of micronutrients may serve as a better alternative due to greater efficiency [17]. When applying micronutrients to the ground, variation in soil characteristics, such as soil pH, drainage and moisture-holding capacity, can significantly impact mobility, solubility and the root uptake of nutrients [15].

The development of improved nutrient guidelines has proven to be beneficial since most of the established guidelines are based on healthy noninfected citrus trees rather than HLB-affected trees [18]. Several studies have examined the effects of various fertilizer types and rates on citrus health and yield [4,9,19,20,21,22]. The fruit yield increased in ‘Valencia’ sweet orange trees (*Citrus × sinensis*) when evaluating varying rates of manganese (Mn) via foliar application at 3× the recommended rate [23]. Additionally, when examining the effect of foliar application rates on mandarins, increased rates of boron (B) and zinc (Zn) above the recommended guidelines resulted in greater fruit yield and quality [24]. The effect of controlled-release fertilizer (CRF) formulations has also been tested on HLB-affected citrus, with CRF treatments resulting in a greater concentration of nitrogen (N), calcium (Ca), sulfur (S) and B in leaves compared to soluble dry granular fertilizers [10]. Furthermore, [25] found that CRF formulations resulted in significantly high yields in HLB-affected sweet orange trees relative to conventional fertilizer sources.

The application of various types and rates of fertilizers may also have subsequent effects on the microbial communities in the soil, notably those that reside within the rhizosphere (a portion of the soil that encompasses the roots of plants). When studying the effects of various N fertilizer treatments on wheat health and rhizosphere composition, Ref. [26] found that the abundance of dominant soil bacteria was significantly altered through changes in pH from long-term fertilization. Similarly, Ref. [27] found that tea orchards treated with a long-term application of organic fertilizer resulted in significant shifts in soil pH correlated with substantial differences in rhizosphere bacterial diversity. Potential shifts in rhizosphere community composition from fertilizer may be crucial for plant health, as they consist of plant-growth-promoting rhizobacteria (PGPR), which can improve plant growth through symbiotic relationships [28]. Some bacteria, such as *Bacillus subtilis*, can assist plant hosts directly in the defense against pathogens [29]. Furthermore, PGPR can make plant essential nutrients available for root uptake, as they are capable of solubilizing nutrients into forms that can be utilized by plants [30]. Further insight into the interactions shared between fertilizer regimens, rhizosphere microbial composition and root health is still needed, specifically toward citrus. This study aimed to determine the impact of foliar and ground-applied fertilizer treatments on grapefruit root health and rhizosphere composition at varying planting densities. It is predicted that grapefruit treated with greater nutrient concentrations will result in a more diverse rhizosphere bacterial community composition.

## 2. Results

### 2.1. Soil Nutrient Concentrations

Soil nutrient concentrations were influenced by planting density treatments, but no clear patterns were established (Table 1). In September 2020, trees planted in a high density (975 trees ha^−1^) had significantly greater soil Mg (20%, Figure 1A), whereas no significant differences were observed for Ca (Figure 2A). In September 2020, trees planted in a high density had significantly greater soil Zn (31%, Figure 3A) than trees planted in a medium density (440 trees ha^−1^). However, trees planted in a medium density had significantly greater soil P (17%, Figure 1A) compared to trees planted in a high density. In January 2021, trees planted in a low density (300 trees ha^−1^) had significantly greater soil Zn (23%, Figure 3A) compared to trees planted in a high density. In May 2021, trees planted in a high density had significantly greater soil Mn (29%, Figure 3A) and Zn (28%, Figure 3A) compared to trees planted in a low density. In September 2021, trees planted in a high density had significantly greater soil P (25%, Figure 1A) and B (14%, Figure 4A) compared to trees planted in a medium density. In January 2022, trees planted in a high density had significantly greater soil Zn (36%, Figure 3A) compared to trees planted in a medium density. 

Ground-applied CRF treatments influenced soil nutrient concentrations; however, similarly to what was reported for planting density, no patterns were established (Table 2). Soil samples fertilized with CRF1 treatment had significantly greater K (41%, Figure 1B) and B (34%, Figure 4B) than those fertilized with the CRF2 treatment in September 2020. However, soil receiving CRF2 treatment had significantly greater Mn (56%, Figure 3B) than the CRF1 treatment. In January 2021, soil receiving CRF2 treatment had significantly greater P (16%, Figure 1B) and Zn (21%, Figure 3B) than those fertilized with CRF1. In May 2021, soil fertilized with the CRF2 treatment had significantly greater Mg (11%, Figure 1B), Mn (75%, Figure 3B), Zn (26%, Figure 3B) and B (23%, Figure 4B) than those fertilized with CRF1. In September 2021, soil fertilized with CRF2 had significantly greater Zn (41%, Figure 3B), Mn (100%, Figure 3B) and Cu (11%, Figure 3B) than those fertilized with CRF2. In January 2022, soil fertilized with CRF2 had significantly greater B (23%, Figure 4B) than soil fertilized with CRF1. No significant differences were detected for Ca (Figure 2B).

As previously reported for planting densities and ground-applied fertilizers, foliar fertilizer treatments affected soil nutrient concentrations. However, no clear trends were observed (Table 3) and no significant differences were detected in soil macronutrients, except for K in September 2020 (Figure 1C). Soil samples from 6× foliar treatments had significantly greater Zn (88% and 39%, Figure 3C) and Mn (99% and 47%, Figure 3C) concentrations than treatments fertilized with 0× and 1.5× foliar spray in September 2020. In January 2021, soil receiving 6× foliar sprays had significantly greater Zn (71% and 32.71%, Figure 3C) and Mn (41% and 26%, Figure 3C) concentrations than 0× and 1.5× foliar treatments. In May 2021, soil receiving 6× foliar sprays had significantly greater Zn (110% and 60%, Figure 3C) concentrations than those fertilized with 0× and 1.5× sprays. In January 2022, soil receiving 6× foliar sprays had significantly greater Zn (110%, 30% and 54%, Figure 3C) concentrations than those fertilized with 0×, 1.5× and 3× foliar sprays. Neither Ca or B showed significant differences (Figure 2C and Figure 4C).

### 2.2. Root Nutrient Concentrations and Size

Root nutrient concentrations and size were significantly affected by planting density. During September 2020, trees planted in a high density resulted in significantly greater total root length (60%, Table 4 and Table 5) than trees planted in a low density. In September 2021, the root concentrations of Mg (52%, Figure 5A) were significantly greater in trees planted in a high density than those planted in a low density. In May 2021, the root concentrations of N (18%, Figure 6A), Mn (73%, Figure 7A) and Zn (59%, Figure 7A) were significantly greater in trees planted in a high density compared to the trees planted in a medium density. Additionally, B was significantly higher in high-density plantings compared to those in a low planting density (45%, Figure 8A). In January 2022, trees planted in a high density had significantly greater Zn (59%, Figure 7A) and Mn (73%, Figure 7A) in their roots than those grown in a medium density. No significant differences were observed in root density in response to planting density (Figure 9A).

Root nutrient concentrations and measurements were significantly affected by ground-applied CRF treatments. In September 2020, grapefruit treated with the CRF2 treatment resulted in roots with a significantly greater root density (40%, Figure 9B) and grapefruit fertilized with the CRF1 treatment was significantly greater than those fertilized with the CRF2. In May 2021, the root concentrations of K (17.32%, Figure 6B) were significantly greater in trees fertilized with the CRF1 treatment compared to the CRF2. However, the root concentrations of P (11%, Figure 5B) and Mg (17%, Figure 5B) were significantly greater in trees fertilized with the CRF2 treatment compared to the CRF1. Additionally, the total root volume (122%, Table 6 and Table 7) and total root area (63%, Table 8 and Table 9) of grapefruit fertilized with the CRF2 treatment were significantly greater than those fertilized with the CRF1. In September 2021, the root concentrations of Mn (52%, Figure 7B) and Zn (35%, Figure 7B) were significantly greater in trees fertilized with CRF2 compared to CRF1. In January 2022, the root concentrations of P (20%, Figure 5B) and Mg (17%, Figure 5B) were significantly greater in trees fertilized with CRF2 compared to CRF1. However, the root concentrations of K (17%, Figure 6B) were significantly greater in trees fertilized with CRF1 compared to CRF2. Ground-applied fertilizer treatments were not observed to have a significant effect on the root B concentrations at all sampling times (Figure 8B). No significant differences were observed in root nutrient concentrations in response to foliar fertilizer treatments (Figure 5C, Figure 6C, Figure 7C and Figure 8C); additionally, no significant difference in root density in response to foliar fertilizer treatments was observed (Figure 9C).

### 2.3. Rhizosphere Microbiome Diversity

Data were log-transformed to reduce error rates caused by rarefaction and later utilized for alpha and beta diversity analyses of the rhizosphere bacterial community through the R package “Phyloseq” v1.24.0 (McMurdie and Holmes 2013). Rhizosphere bacterial alpha diversity did vary according to treatments according to the Shannon index. Grapefruit trees treated with CRF 2 had a greater bacterial alpha diversity than those treated with CRF 1 (Figure 10A). Grapefruit trees treated with 3× foliar fertilizer had a greater bacterial alpha diversity compared to those treated with 0×, 1.5× and 6× foliar fertilizer (Figure 11A).

Beta diversity analyses included principal coordinate analysis (PCoA) on Bray–Curtis distances. An ANOSIM test was performed to determine significant differences in beta diversity between treatments. Rhizosphere bacterial beta diversity did vary according to treatments according to the Shannon index. Grapefruit trees treated with CRF 2 had greater bacterial beta diversity than those treated with CRF 1 (Figure 10B). Grapefruit trees treated with 3× foliar fertilizer had greater bacterial beta diversity compared to those treated with 0×, 1.5× and 6× foliar (Figure 11B).

There was variation in the relative abundance of bacterial taxonomic orders among treatments. Grapefruit trees treated with 0× foliar fertilizer had a rhizosphere bacterial community with a significantly greater abundance of Reyranellales (*p* < 0.05), Rhizobiales (*p* < 0.05) and Rickettsiales (*p* < 0.05) compared to those from the 1.5× foliar treatment. Additionally, grapefruit trees at 0× foliar fertilizer had a rhizosphere bacterial community with a significantly greater abundance of Nitrosotaleales (*p*  <  0.05), Vicinamibacterales (*p* < 0.05), Tistrellales (*p* < 0.05) and Solirubrobacterales (*p*  <  0.05) compared to those from the 3× foliar fertilizer treatment. Furthermore, grapefruit trees with 0× foliar fertilizer had a rhizosphere bacterial community with a significantly greater abundance of Nitrososphaerales (*p* < 0.05), Clostridiales (*p*  <  0.05), Caulobacterales (*p*  <  0.05) and Rickettsiales (*p*  <  0.05) compared to 6× foliar treatment.

## 3. Discussion

Planting density affected root nutrient concentrations and root size, with notably significantly greater root concentrations of Mn and Zn concentrations in a high density than in a medium density in May 2021 and January 2022. When planting grapefruit in higher densities, the greater presence of roots within a given soil area may have allowed for greater root interception of essential nutrients. Similarly, Gezahegn et al. found that closely spaced fava bean plants had increased root elongation, increasing moisture and nutrient uptake [31]. Additionally, the greater presence and activity of grapefruit roots from high-density plantings may also increase the amount of root exudates in these soils, potentially contributing to observed increases in root concentrations of Mn and Zn. Root exudates function to recruit microbes from the bulk soil to the rhizosphere for a multitude of tasks, whereby some of which range from plant defense to nutrient acquisition [32,33]. The type and amount of root exudates released vary according to the plant’s growth stage and the root characteristics, such as root physiology and morphology [34,35].

Changes in root nutrient concentrations and measured parameters were more influenced by ground-applied CRF treatments than foliar fertilizer treatments, due to application sites being closer to the root zone than foliar application sites in the canopy. Long-term ground-applied CRF use can have more pronounced effects on soil characteristics, such as pH (commonly increased soil acidification), when compared to foliar fertilizers, affecting the availability of plant essential nutrients [36,37]. Furthermore, long-term excessive fertilizer applications have also been shown to alter total organic carbon (TOC), basic cation content and soil physical properties, further emphasizing the effect of ground-applied CRF on overall root health [38,39]. The phenology of citrus trees with several vegetive flushes during the year, in combination with the sub-tropical weather of Florida (with an abundance of rain during the summer and fall seasons) and the deterioration in tree health caused by HLB disease, may have contributed toward the inconsistency in nutrient concentrations, and the lack of observed patterns during the time periods of the study.

When comparing the impact of ground-applied CRF on root health, the CRF2 treatment had an overall greater beneficial effect compared to the CRF1 treatment, notably in May 2021, with significant increases in the total root area and total root volume of grapefruit. Similarly, [40] found that 2× the dose of micronutrients (Mn, Zn and B) in HLB-affected sweet orange trees led to a higher median root lifespan. Greater concentrations of micronutrients (3× B, Fe, Mn and Zn) in the CRF2 treatment may have also contributed toward better root health. Nutrient supply is imperative to disease control because nutrients influence plant resistance, pathogen vigor, growth and associated factors [15].

Both foliar and ground-applied fertilizer treatments had a greater impact on soil micronutrient concentrations compared to planting density at all timepoints, as confirmed through Pearson correlation coefficients. This was expected, as micronutrients were the main component that changed through the foliar and ground-applied fertilizer treatments.

Excess amounts of micronutrients provided by the CRF2 treatment may have improved HLB-affected root health by reducing the activity of pathogenic organisms, such as *C*Las. Greater concentrations of micronutrients, such as Mn and Zn, have been shown to exhibit antimicrobial functions [41]. For instance, Zn has been shown to exhibit host-pathogen interactions, as increased concentrations have been shown to suppress the growth of potential phytopathogens [42,43], thus promoting soil and plant health. Higher doses of foliar Mn in HLB-affected trees reduced symptom severity [16].

Significantly greater abundances of Rhizobiales and Vicinamibacterales may have been recruited from excess root exudates released from stressed HLB-affected grapefruit treated with 0× UF/IFAS foliar fertilizer compared to the other trees treated with higher doses of foliar fertilizers. Rhizobiales are a bacterial order of interest as they can interact with host plants to produce auxins, vitamins and N fixation, and protect the plants against stress [44]. Additionally, Vicinamibacterales have also been classified as PGPR, as they have been associated with increased plant available nutrient concentrations (specifically N and P) in rice rhizosphere soil [45]. Typically, greater amounts of exudates are released from roots during periods of stress for the purpose of recruiting microorganisms, specifically plant-growth-promoting rhizobacteria (PGPR) to assist in the acquisition of plant essential resources that would otherwise be unavailable [46,47,48]. Similarly, host plants grown under nutrient-deficient conditions have been shown to increase the synthesis of a root exudate known as Strigolactones to promote mycorrhizal fungal recruitment for nutrient acquisition [49,50]. Moreover, flavonoids released from the roots of nutrient-deficient legumes have been shown to stimulate bacterial root infection, leading to the establishment of nodules that promote N fixation [51].

## 4. Materials and Methods

### 4.1. Experimental Design

The study site is located at the University of Florida Institute of Food and Agricultural Sciences (UF/IFAS) Indian River Research and Education Center in Fort Pierce, FL (latitude 27.435342°, longitude -80.445197°, altitude 10 m). Plant material consisted of ‘Ray Ruby’ grapefruit trees (*Citrus × paradisi*) grafted on ‘Kuharske’ citrange (*Citrus × sinensis × Citrus trifoliata*). Trees were planted in September 2013 in flatwood soils (Pineda sands classified as loamy, siliceous, active, hyperthermic Arenic Glossaqualfs) which are poorly drained and consist of 96% sand, 2.5% silt and 1.5% clay and have an argillic soil layer at 90 cm below the soil surface. The average soil pH was 5.8 and the cation exchange capacity (CEC) was 3.5 c_mol_ kg^−1^. Trees were grown on raised beds roughly 1 m tall to facilitate drainage, with swales 15 m between beds. Irrigation was delivered using 39.7 L h^−1^ microjet sprinklers (Maxijet, Dundee, FL, USA). The original experiment, which only focused on aboveground parameters, has been described in [9] and was arranged in a split-split-plot design with three factors, each consisting of plant densities, ground-applied controlled-release fertilizers (applied in February, May and September) and foliar-applied fertilizer combinations (applied in March, June and October). The three planting densities implemented were low (300 trees ha^−1^), medium (440 trees ha^−1^) and high (975 trees ha^−1^). The study consisted of two ground-applied fertilizer treatments, specifically, controlled-release fertilizer blends 1 (CRF1): 12.00N-1.31P-11.62K and micronutrients at 1× the UF/IFAS recommendation with micronutrients as sulfates (12-3-14 1× Micro) and CRF2: enhanced 12.00N-1.31P-11.62K with 2× Mg and 2.5× the UF/IFAS recommendation with micronutrients as sulfur-coated products (#12-3-14 2.5× Micro, Table 10 and Table 11). Additionally, there were four foliar fertilizer treatments, which included 0×, 1.5×, 3× and 6× the UF/IFAS recommendation [14]. Root and soil parameter sampling was performed every 4 months from September 2020 to January 2022. Rhizosphere samples were collected in January 2021.

### 4.2. Soil Nutrient Analysis 

A soil auger (One-Piece Auger model #400.48, AMS, Inc., American Falls, ID) 7 cm in diameter and 10 cm in depth was used to collect soil samples for nutrient concentrations. A single core was taken within the irrigated zone from one tree per experimental plot (total of 96) across eight experimental blocks. Soil samples were analyzed for extractable N, P, K, Mg, Ca, S, B, Zn, Mn, Fe and Cu. 

Soil samples were dried overnight at 80 ℃, and nutrient concentrations were determined using Mehlich III extraction [52]. A total of 25 mL of Mehlich III extractant solution (0.2 M CH_3_COOH + 0.015 M NH_4_F + 0.013 M HNO_3_ + 0.001 M EDTA + 0.25 M H_4_NO_3_) was pipetted into extraction tubes containing 2.5 ± 0.05 g of soil. Soil nutrient concentrations were measured using inductively coupled plasma optical emission spectroscopy (ICP-OES, Spectro Ciros CCD, Fitzburg, MA, USA) [14].

### 4.3. Root Parameter Analysis

Root parameters were measured using a minirhizotron system (CID Bio-Science CI-602, CID Bioscience, Inc. Camas, WA, USA). Each tube consisted of three scannable windows at different depths (0–19 cm, 19–39 cm and 39–59 cm), and all three windows in every tube were scanned for each sampling point. After images were acquired, average root length and density were calculated using commercial software (RootSnap™ Version 1.3.2.25, CID Bio-Science, Camas, WA, USA).

### 4.4. Deoxyribonucleic Acid (DNA) Isolation and Quantification

Rhizosphere samples were taken shortly after bulk soil sampling, which consisted of rhizosphere soil (located around the roots) being lightly shaken from the roots and placed in 50 mL sterile tubes. Approximately, 50 g of the soil was collected and stored at −20 °C before DNA extraction. Approximately, 15 mL of 1× sterile phosphate-buffered saline (800 mL distilled water, 8 g NaCl, 0.2 g KCl, 1.44 g Na_2_PO_4_ and 0.24 g KH_2_PO_4_) was added to the sample and shaken by hand for 15 s. Roots were removed with forceps and discarded, the remaining soil was centrifuged at 3000 g for 15 min and the supernatant was discarded. Soil DNA was extracted from 0.25 g of the soil pellet using the DNeasy PowerSoil Kit (Qiagen Inc., Germantown, MD, USA) according to the manufacturer’s instructions. A fluorometer (Qubit, Thermofisher Scientific, Wilmington, NC, USA) was used to quantify the extracted DNA and determine whether the DNA was concentrated enough for sequencing (>1 ng/µL). Rhizosphere DNA was amplified with 515Fa/926R universal bacterial [53] primers and sequenced at the Genomics and Microbiome Core Facility at Rush University, Chicago, IL, USA. 

### 4.5. Rhizosphere Microbiome Diversity and Statistical Analysis 

After sequencing, bioinformatic data were processed using DADA2 [54] within the Qiime 2 [55] package. Raw sequences were demultiplexed. DADA2 was used to filter chimeras, primers and adapters and assemble pair-ended sequences. Taxonomy was assigned to amplicon sequence variants (ASVs) with the reference dataset SILVA 128 database for 16S rRNA using a naïve Bayes classifier in Qiime2 [56]. Alpha and beta diversity analyses of the bacterial community were performed on log-normalized data to avoid an increase in error rates due to rarefaction [57] with the R package “Phyloseq” v1.24.0 [58]. Alpha diversity analyses included the number of observed ASVs (Shannon index), whereas beta diversity analyses included principal coordinate analysis (PCoA) on weighted UniFrac distances. A two-way analysis of similarities (ANOSIM) test was performed to determine significant differences in beta diversity between treatments.

### 4.6. Plant and Soil Data Statistical Analysis 

An analysis of variance (three-way ANOVA) was performed using statistical software (R Version 3.6.0, RStudio, Boston, MA, USA). The main effect means were separated using Tukey’s honestly significant difference post hoc test. Differences were considered to be significant when *p*-values were less than or equal to 0.05. Additionally, Pearson correlation coefficients were computed.

## 5. Conclusions

In this study, we examined the impact of planting densities, ground-applied controlled-release fertilizers and foliar fertilizer dosages on both soil and plant parameters, including rhizosphere community composition. The use of the CRF2 ground-applied fertilizer resulted in higher soil nutrient concentrations through all time points of the study, notably with significant differences in Zn and Mn. Additionally, increased micronutrient concentrations provided by the CRF2 ground-applied fertilizer may have provided additional nutrients required to assist the root health of HLB-affected grapefruit trees. Furthermore, HLB-affected grapefruit at 0× the UF/IFAS recommended foliar fertilizer resulted in a significantly greater abundance of Vicinamibacterales and Rhizobiales, whereby both of which are PGPR that may have been recruited to assist grapefruit health under nutrient-deprived conditions.

## Figures and Tables

**Figure 1 plants-12-01659-f001:**
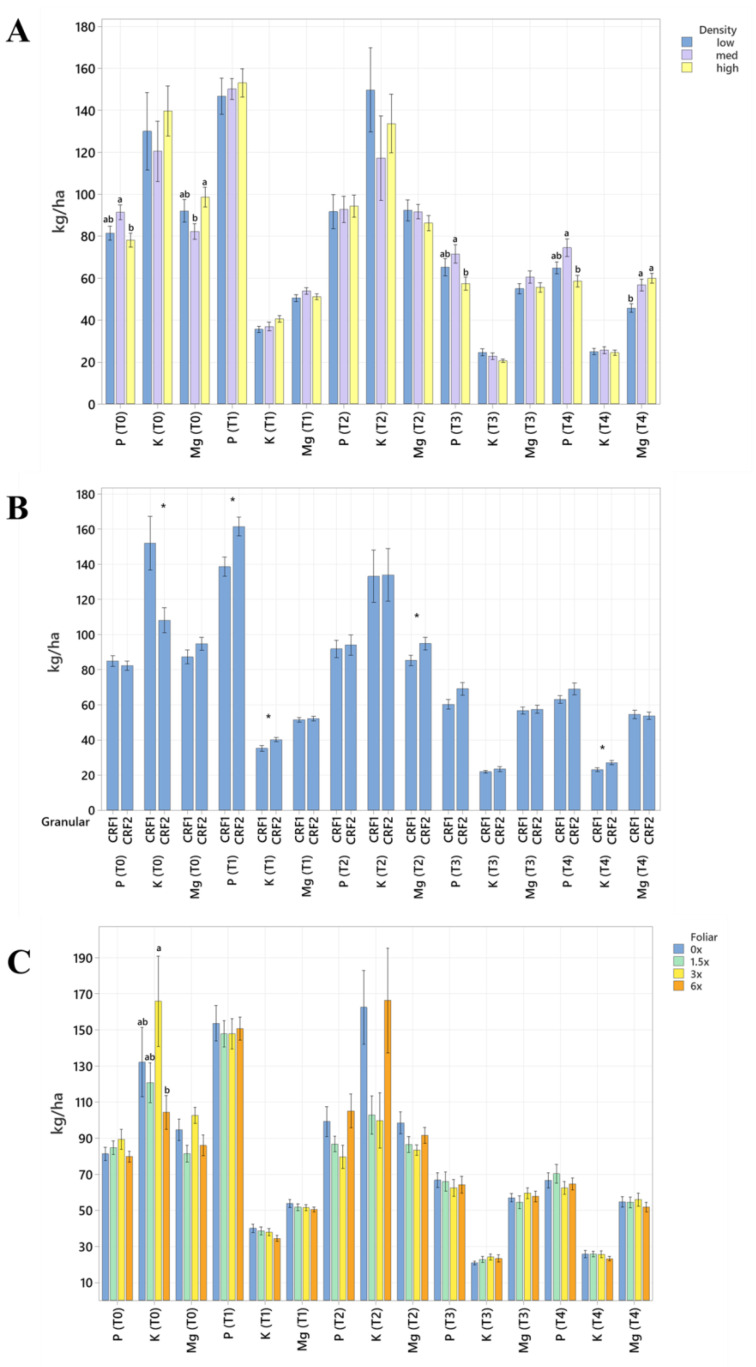
Soil macronutrient concentrations of ‘Ray Ruby’ grapefruit grafted on ‘Kuharske’ citrange planted in flatwood soils located in Fort Pierce, FL, USA, in response to (**A**) planting density, (**B**) ground-applied fertilizer and (**C**) foliar fertilizer treatments during September 2020 (T0), January 2021 (T1), May 2021 (T2), September 2021 (T3) and January 2022 (T4). Bars are ± standard deviation of the mean. Treatments with * and different letters were considered to be significantly different (*p* < 0.05).

**Figure 2 plants-12-01659-f002:**
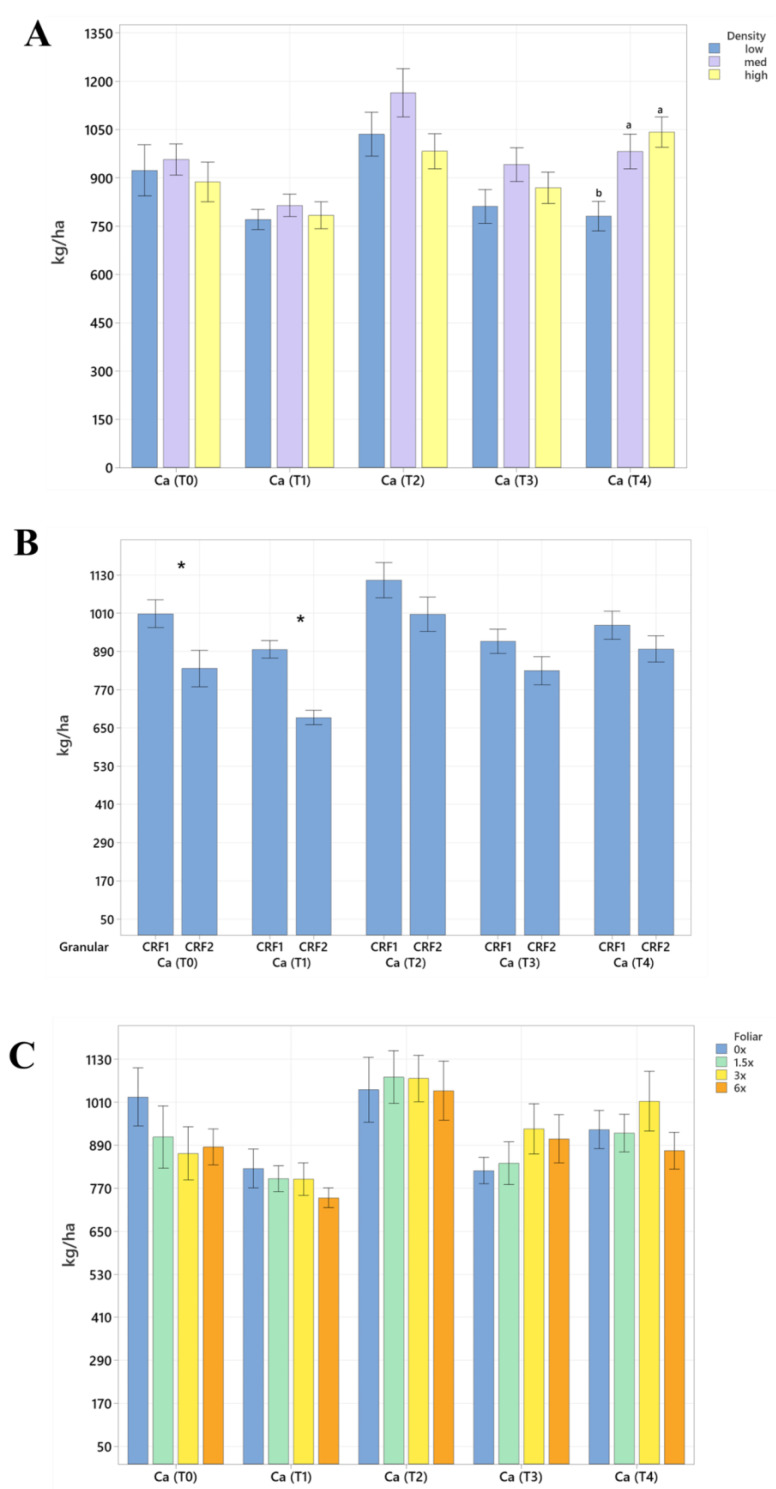
Soil Ca concentrations of ‘Ray Ruby’ grapefruit grafted on ‘Kuharske’ citrange planted in flatwood soils located in Fort Pierce, FL, USA, in response to (**A**) planting density, (**B**) ground-applied fertilizer and (**C**) foliar fertilizer treatments during September 2020 (T0), January 2021 (T1), May 2021 (T2), September 2021 (T3) and January 2022 (T4). Bars are ± standard deviation of the mean. Treatments with * and different letters were considered to be significantly different (*p* < 0.05).

**Figure 3 plants-12-01659-f003:**
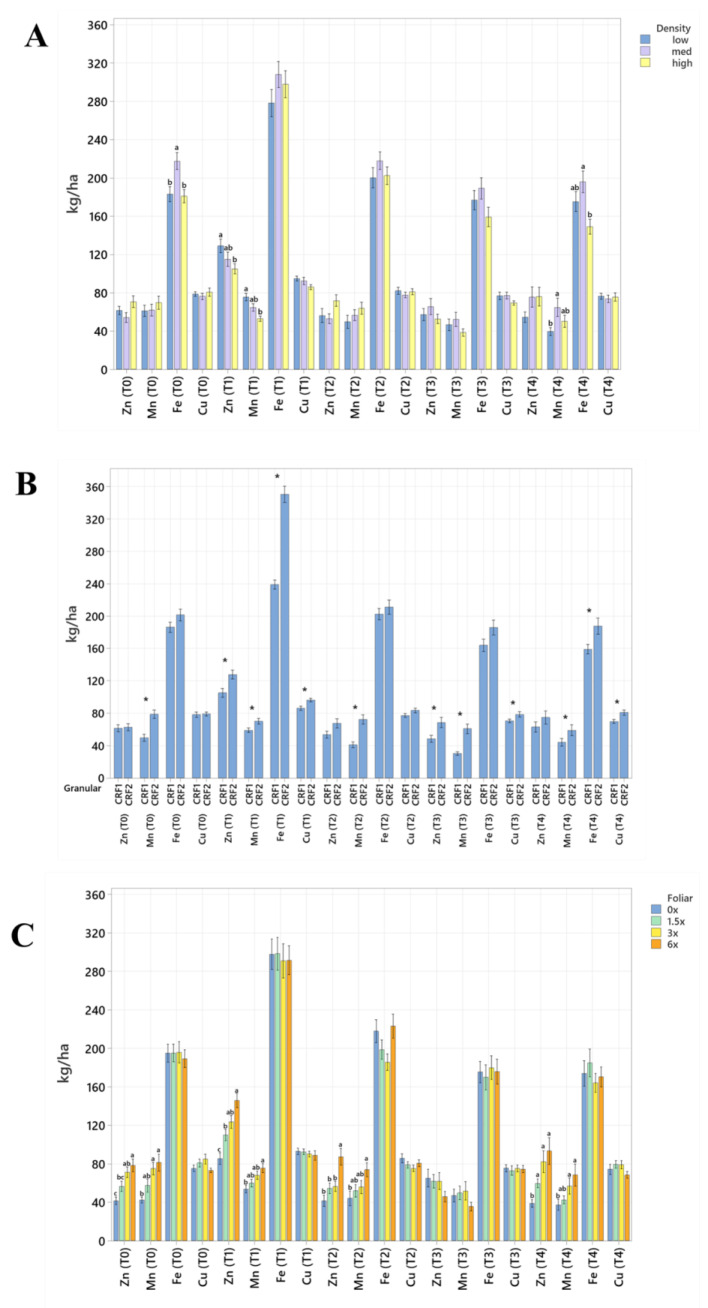
Soil micronutrient concentrations of ‘Ray Ruby’ grapefruit grafted on ‘Kuharske’ citrange planted in flatwood soils located in Fort Pierce, FL, USA, in response to (**A**) planting density, (**B**) ground-applied fertilizer and (**C**) foliar fertilizer treatments during September 2020 (T0), January 2021 (T1), May 2021 (T2), September 2021 (T3) and January 2022 (T4). Bars are ± standard deviation of the mean. Treatments with * and different letters were considered to be significantly different (*p* < 0.05).

**Figure 4 plants-12-01659-f004:**
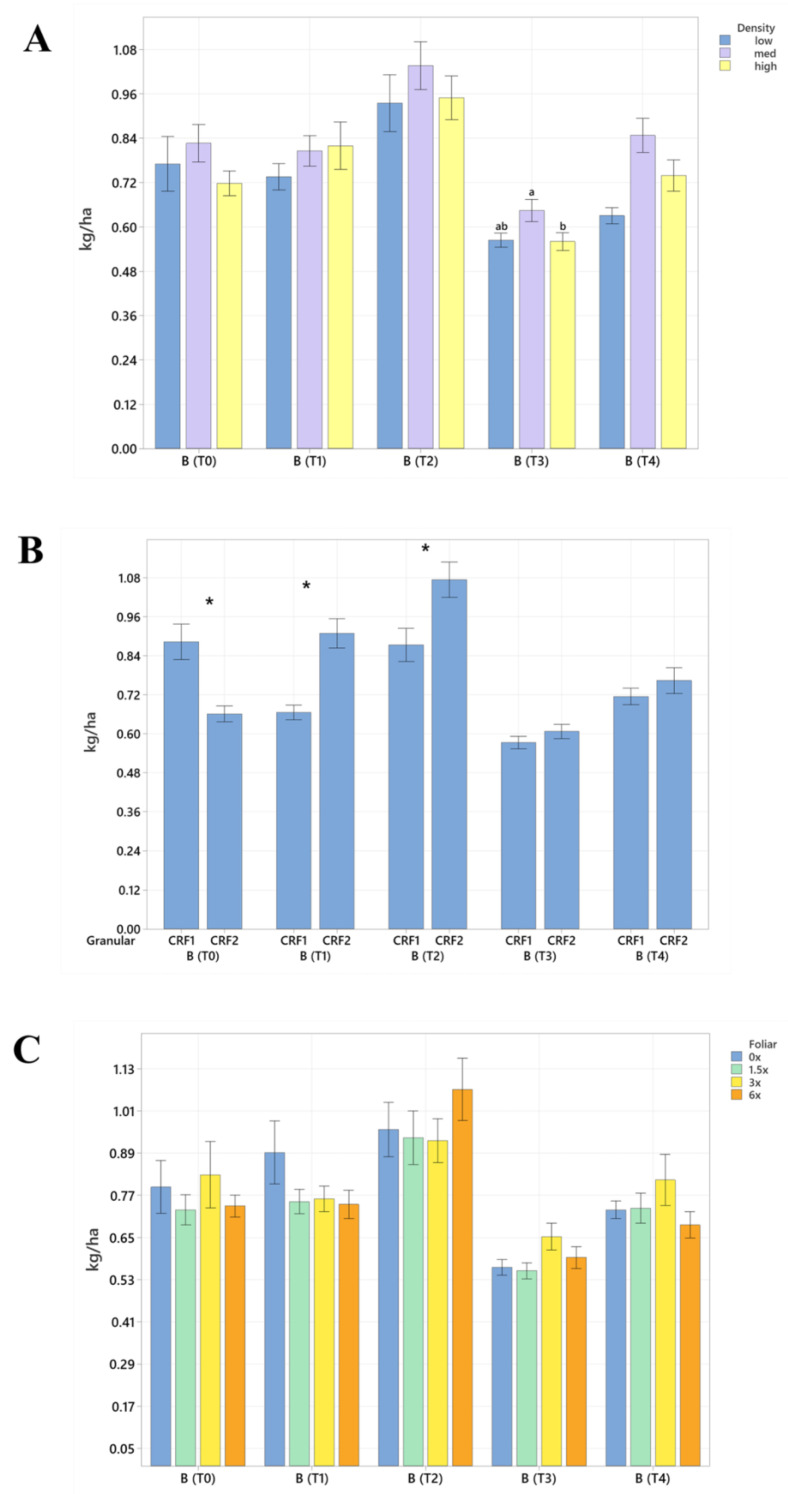
Soil boron concentrations of ‘Ray Ruby’ grapefruit grafted on ‘Kuharske’ citrange planted in flatwood soils located in Fort Pierce, FL, USA, in response to (**A**) planting density, (**B**) ground-applied fertilizer and (**C**) foliar fertilizer treatments during September 2020 (T0), January 2021 (T1), May 2021 (T2), September 2021 (T3) and January 2022 (T4). Bars are ± standard deviation of the mean. Treatments with * and different letters were considered to be significantly different (*p* < 0.05).

**Figure 5 plants-12-01659-f005:**
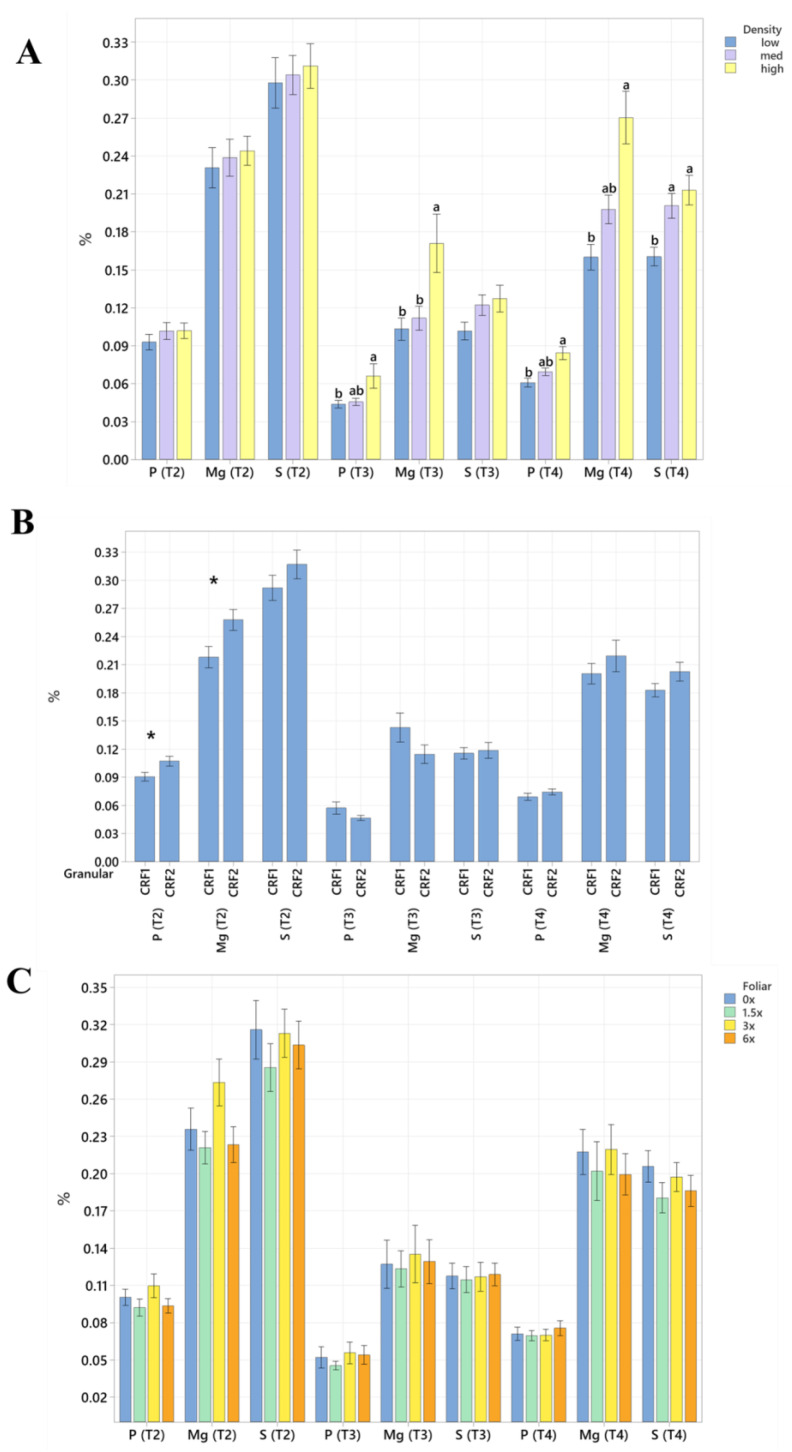
Root P, Mg and S concentrations of ‘Ray Ruby’ grapefruit grafted on ‘Kuharske’ citrange planted in flatwood soils located in Fort Pierce, FL, USA. Graphs indicate response to (**A**) planting density, (**B**) ground-applied fertilizer and (**C**) foliar fertilizer treatments during May 2021 (T2), September 2021 (T3) and January 2022 (T4). Bars are ± standard deviation of the mean. Treatments with * and different letters were considered to be significantly different (*p* < 0.05).

**Figure 6 plants-12-01659-f006:**
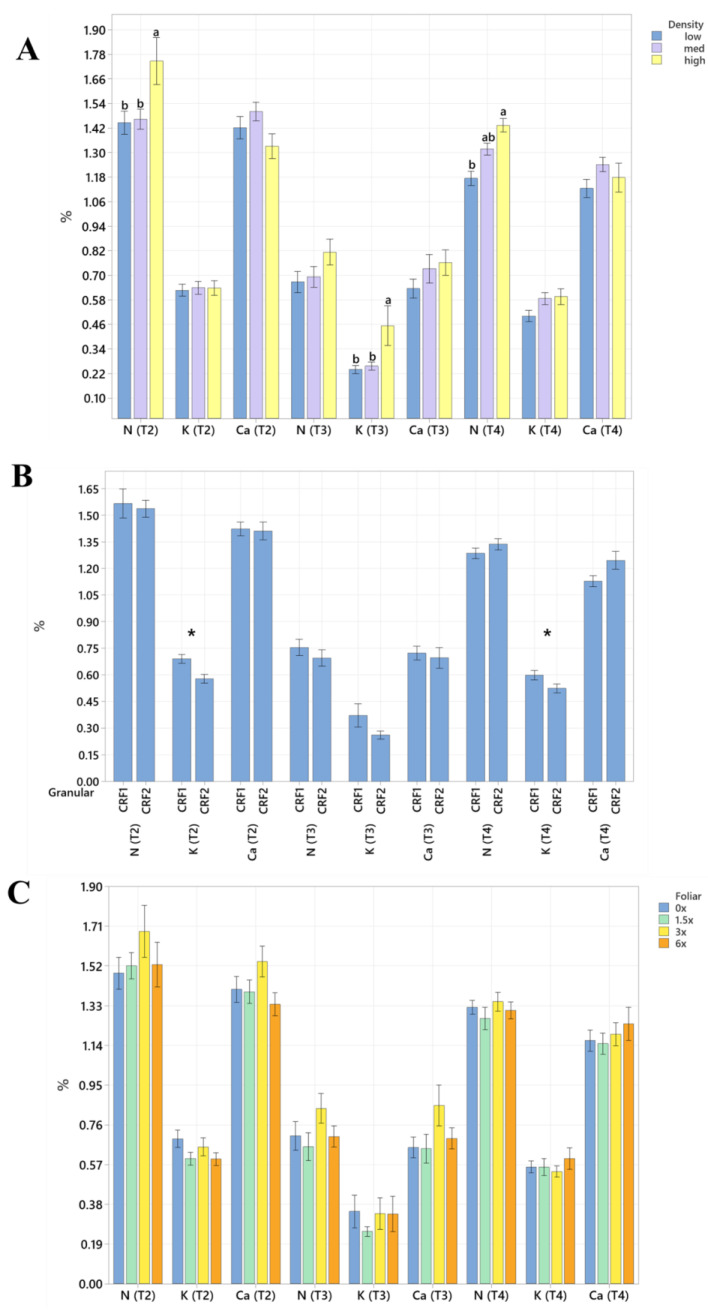
Root N, K and Ca concentrations of ‘Ray Ruby’ grapefruit grafted on ‘Kuharske’ citrange planted in flatwood soils located in Fort Pierce, FL, USA. Graphs indicate response to (**A**) planting density, (**B**) ground-applied fertilizer and (**C**) foliar fertilizer treatments during May 2021 (T2), September 2021 (T3) and January 2022 (T4). Bars are ± standard deviation of the mean. Treatments with * and different letters were considered to be significantly different (*p* < 0.05).

**Figure 7 plants-12-01659-f007:**
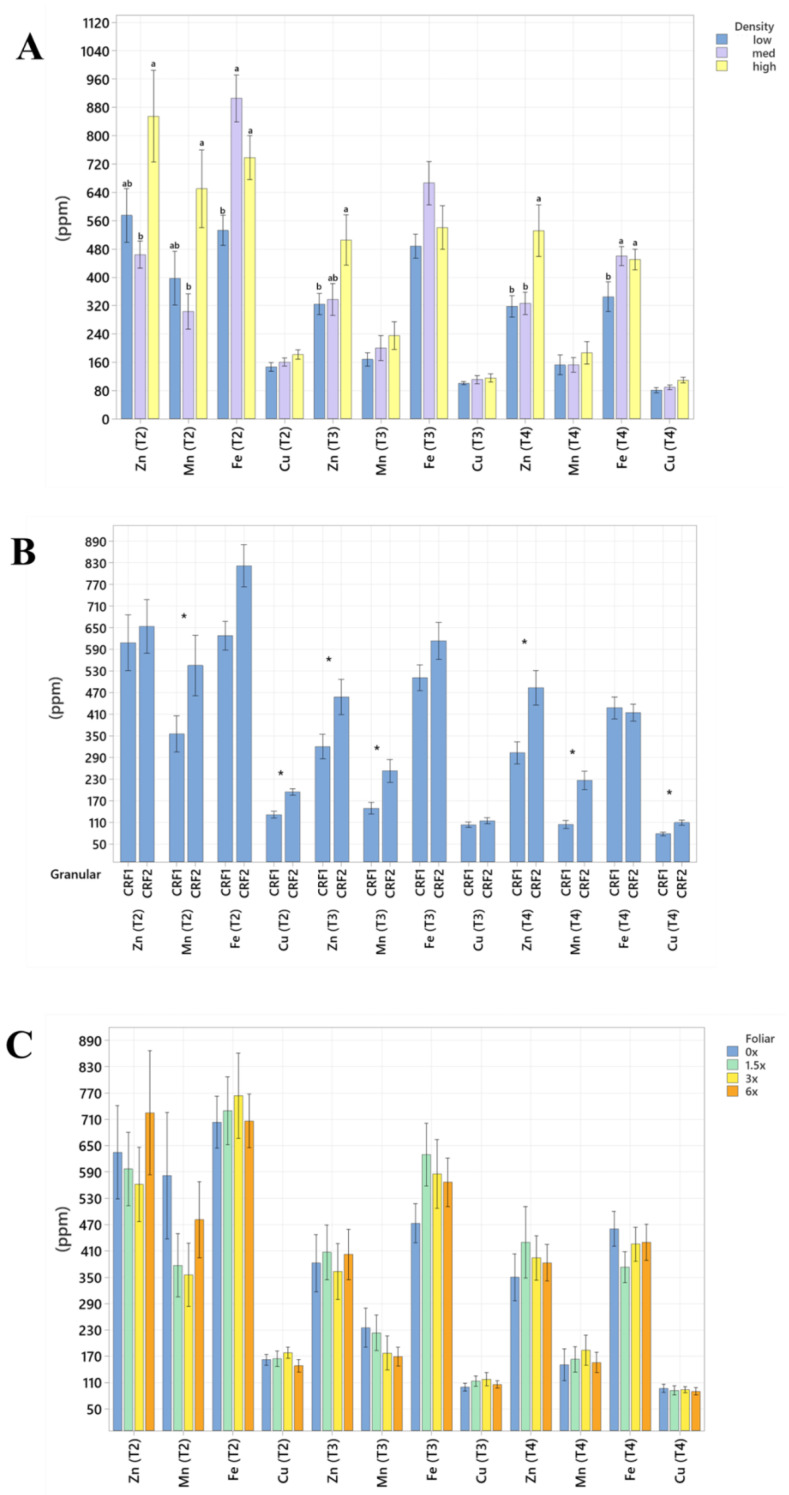
Root micronutrient concentrations of ‘Ray Ruby’ grapefruit grafted on ‘Kuharske’ citrange planted in flatwood soils located in Fort Pierce, FL, USA. Graphs indicate response to (**A**) planting density, (**B**) ground-applied fertilizer and (**C**) foliar fertilizer treatments during May 2021 (T2), September 2021 (T3) and January 2022 (T4). Bars are ± standard deviation of the mean, and treatments with * and different letters were considered to be significantly different (*p* < 0.05).

**Figure 8 plants-12-01659-f008:**
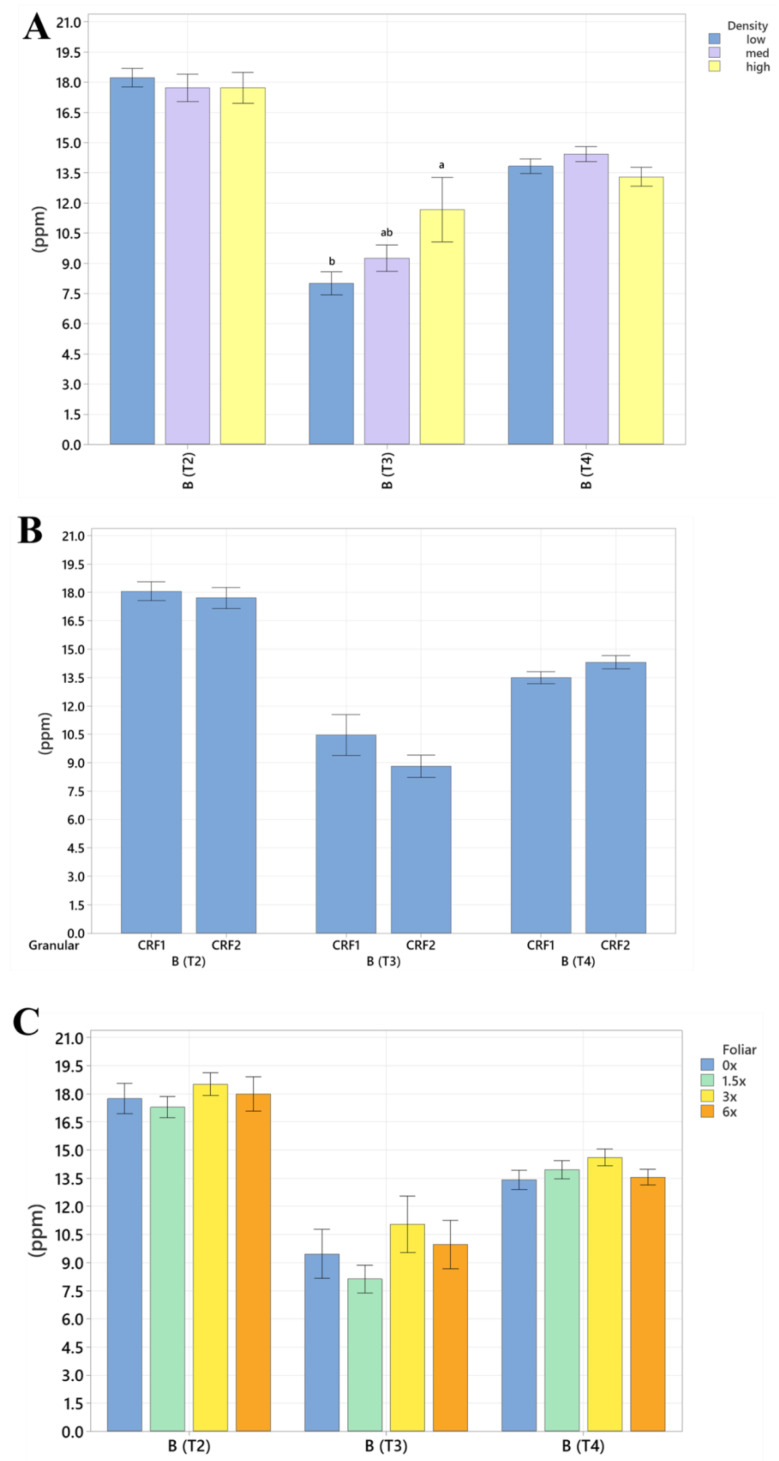
Root B concentrations of ‘Ray Ruby’ grapefruit grafted on ‘Kuharske’ citrange planted in flatwood soils located in Fort Pierce, FL, USA. Graphs indicate response to (**A**) planting density, (**B**) ground-applied fertilizer and (**C**) foliar fertilizer treatments during May 2021 (T2), September 2021 (T3) and January 2022 (T4). Bars are ± standard deviation of the mean, and treatments with different letters were considered to be significantly different (*p* < 0.05).

**Figure 9 plants-12-01659-f009:**
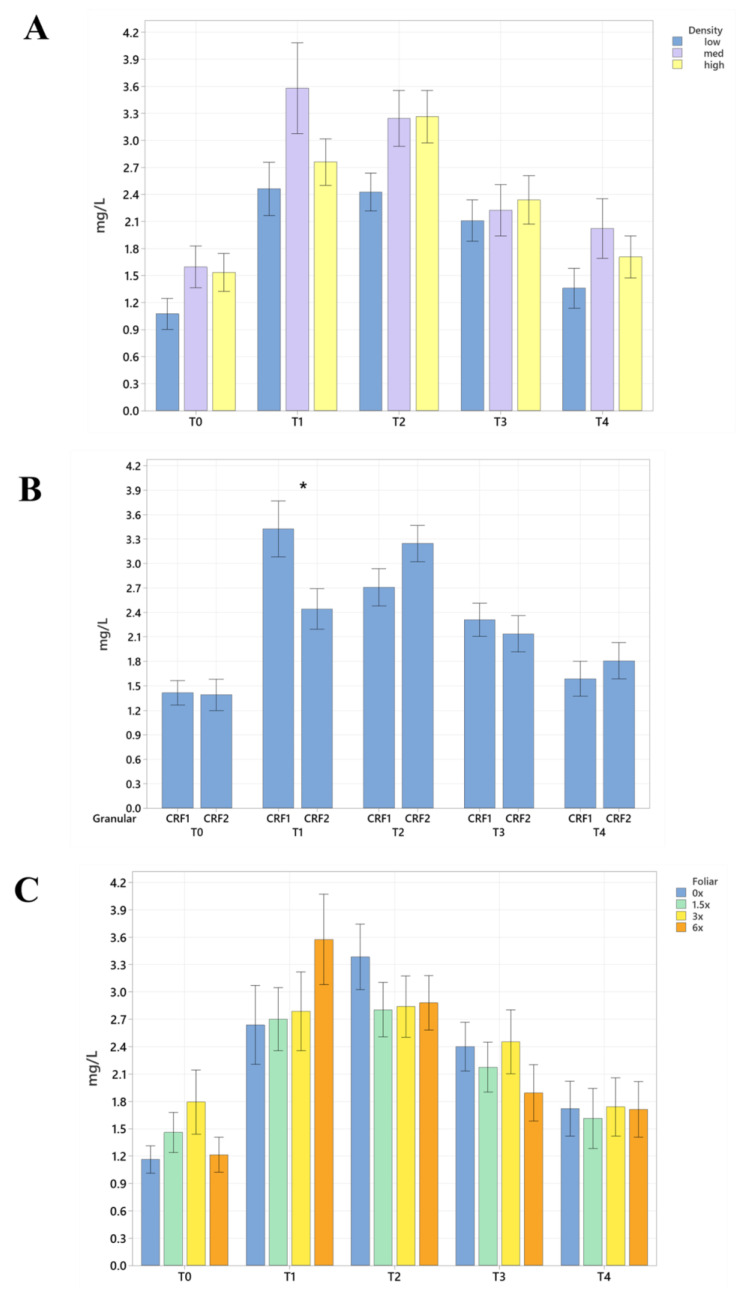
Root density of ‘Ray Ruby’ grapefruit grafted on ‘Kuharske’ citrange planted in flatwood soils located in Fort Pierce, FL, USA, in response to (**A**) planting density, (**B**) ground-applied fertilizer and (**C**) foliar fertilizer treatments during September 2020 (T0), January 2021 (T1), May 2021 (T2), September 2021 (T3) and January 2022 (T4). Bars are ± standard deviation of the mean, and treatments with * were considered to be significantly different (*p* < 0.05).

**Figure 10 plants-12-01659-f010:**
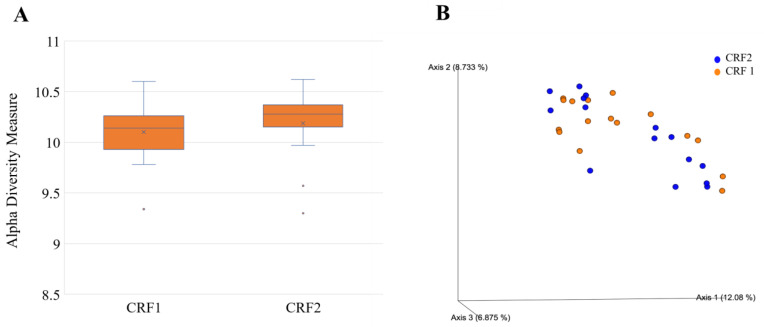
Alpha (**A**) and beta diversity (**B**) in rhizosphere bacteria of ‘Ray Ruby’ grapefruit grafted on ‘Kuharske’ citrange planted in flatwood soils located in Fort Pierce, FL, USA, and grown with different ground-applied fertilizer treatments. Alpha diversity was measured using the Shannon index of rhizosphere bacteria among treatments. Plotted in Figure (**A**) are boxes (interquartile), the median (line within each box), the mean (× within each box), and whiskers (lowest and greatest values). Principal coordinates analysis (PCoA) based on the Bray–Curtis dissimilarity matrix of rhizosphere bacterial samples can be found in Figure (**B**), where colors indicate treatment and include CRF1 (orange) and CRF2 (blue).

**Figure 11 plants-12-01659-f011:**
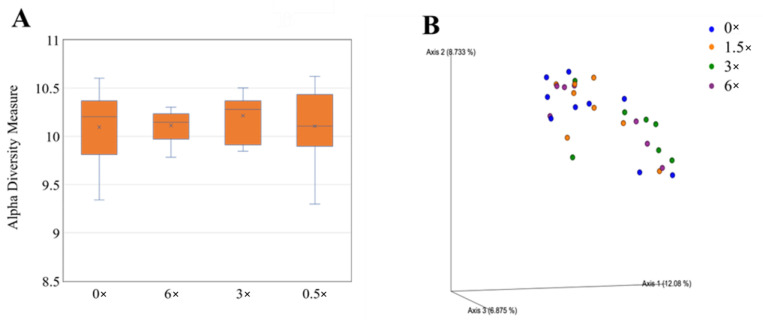
Alpha (**A**) and beta diversity (**B**) in rhizosphere bacteria of ‘Ray Ruby’ grapefruit grafted on ‘Kuharske’ citrange planted in flatwood soils located in Fort Pierce, FL, USA, and treated with various doses of foliar fertilizer applications. Alpha diversity was measured using the Shannon index of rhizosphere bacteria among treatments. Plotted in Figure (**A**) are boxes (interquartile), the median (line within each box), the mean (× within each box), and whiskers (lowest and greatest values). Principal coordinates analysis (PCoA) based on the Bray–Curtis dissimilarity matrix of rhizosphere bacterial samples can be found in Figure (**B**), where colors indicate treatment and include 0× (blue), 1.5× (orange), 3× (green) and 6× (purple) the UF/IFAS recommended.

**Table 1 plants-12-01659-t001:** Corresponding *p*-values and r values (Pearson’s coefficient) of correlations examined between soil nutrient concentrations and planting density treatments from all time points of sampling, which include September 2020, January 2021, May 2021, September 2021 and January 2022.

Nutrient	r	*p*-Value
P	−0.05	0.4
K	0.03	0.25
Mg	−0.01	0.98
Ca	0.12	0.006
B	0.08	0.08
Zn	−0.06	0.18
Mn	0.06	0.9
Fe	0.12	0.08
Cu	0.01	0.79

**Table 2 plants-12-01659-t002:** Corresponding *p*-values and r values (Pearson’s coefficient) of correlations examined between soil nutrient concentrations and ground-applied CRF treatments from all time points of sampling, which include September 2020, January 2021, May 2021, September 2021 and January 2022.

Nutrient	r	*p*-Value
K	−0.05	0.33
P	0.09	0.08
Mg	0.1	0.03
Ca	−0.21	5.12 × 10^−6^
B	0.13	0.004
Zn	0.17	0.0002
Mn	0.4	2.20 × 10^−16^
Fe	0.23	3.17 × 10^−7^
Cu	0.17	0.0002

**Table 3 plants-12-01659-t003:** Corresponding *p*-values and r values (Pearson’s coefficient) of correlations examined between soil nutrient concentrations and foliar fertilizer treatments from all time points of sampling, which include September 2020, January 2021, May 2021, September 2021 and January 2022.

Nutrient	r	*p*-Value
K	−0.03	0.94
P	0.01	0.98
Mg	−0.03	0.39
Ca	−0.02	0.65
B	0.02	0.56
Zn	0.29	9.76 × 10^−11^
Mn	0.24	9.19 × 10^−8^
Fe	−0.09	0.84
Cu	−0.06	0.13

**Table 4 plants-12-01659-t004:** Total root length of ‘Ray Ruby’ grapefruit grafted on ‘Kuharske’ citrange planted in flatwood soils located in Fort Pierce, FL, USA, and treated with ground-applied fertilizer, foliar fertilizer and different planting densities from September 2020 to January 2022.

Root Length (mm)		Sept. 2020	Jan. 2021	May 2021	Sept. 2021	Jan. 2022
Ground-applied ^1^	CRF1	489 ± 120	622.6 ± 161	653.7 ± 143	545.5 ± 112	706.0 ± 95
	CRF2	572 ± 90	432.9 ± 93	450.2 ± 103	509.0 ± 79	655.2 ± 111
Foliar ^2^	0×	504 ± 180	343.2 ± 143	539.3 ± 194	469.7 ± 118	617.5 ± 121
	1.5×	476 ± 101	431.0 ± 78	456.0 ± 100	405.8 ± 95	600.5 ± 124
	3×	401 ± 121	406.5 ± 131	329.8 ± 104	523.1 ± 155	685.1 ± 171
	6×	772 ± 194	1120.8 ± 318	994.8 ± 298	726.6 ± 171	817.8 ± 173
Planting density ^3^	High	738 ± 128	651.6 ± 142	655.2 ± 183	729.2 ± 149	790.8 ± 156
	Low	398 ± 144	529.0 ± 227	559.2 ± 194	382.3 ± 79	508.6 ± 68
	Medium	402 ± 72	442.5 ± 176	593.5 ± 202	495.6 ± 113	747.8 ± 137

^1^ Controlled release fertilizer blends 1 (CRF1): 12N-1.31 P-11.62K and micronutrients at 1× the UF/IFAS recommendation with micronutrients as sulfates (12-3-14 1× Micro) and CRF2: enhanced 12N-1.31 P-11.62K with 2× Mg and 2.5× the UF/IFAS recommendation with micronutrients as sulfur-coated products (#12-3-14 2.5× Micro). ^2^ Foliar fertilizer treatments, which included 0×, 1.5×, 3× and 6× the UF/IFAS recommendation. ^3^ Low (300 trees ha^−1^), medium (440 trees ha^−1^) and high (975 trees ha^−1^).

**Table 5 plants-12-01659-t005:** *p*-values obtained via a three-way ANOVA followed by Tukey’s post hoc test, corresponding to the total root length of ‘Ray Ruby’ grapefruit grafted on ‘Kuharske’ citrange planted in flatwood soils located in Fort Pierce, FL, USA, and treated with ground-applied fertilizer, foliar fertilizer and different planting densities from September 2020 to January 2022. Bold values were considered to be significantly different (*p* < 0.05).

Root Length *p* Values		Sept. 2020	Jan. 2021	May 2021	Sept. 2021	Jan. 2022
Ground-applied ^1^	CRF1—CRF2	0.28	0.74	0.35	0.78	0.75
Foliar ^2^	1.5×–0×	0.84	0.44	0.90	0.98	0.99
	3×–0×	0.99	0.58	0.99	0.99	0.98
	6×–0×	0.28	0.98	0.99	0.52	0.76
	3×–1.5×	0.72	0.99	0.92	0.91	0.97
	6×–1.5×	0.72	0.71	0.85	0.32	0.73
	6×–3×	0.18	0.82	0.99	0.71	0.92
Planting density ^3^	Low—High	0.04	0.77	0.93	0.06	0.27
	Med—High	0.32	0.90	0.86	0.22	0.94
	Med—Low	0.62	0.96	0.66	0.79	0.42

^1^ Controlled-release fertilizer blends 1 (CRF1): 12N-1.31 P-11.62K and micronutrients at 1× the UF/IFAS recommendation with micronutrients as sulfates (12-3-14 1× Micro) and CRF2: enhanced 12N-1.31 P-11.62K with 2× Mg and 2.5× the UF/IFAS recommendation with micronutrients as sulfur-coated products (#12-3-14 2.5× Micro). ^2^ Foliar fertilizer treatments, which included 0×, 1.5×, 3× and 6× the UF/IFAS recommendation. ^3^ Low (300 trees ha^−1^), medium (440 trees ha−1) and high (975 trees ha^−1^).

**Table 6 plants-12-01659-t006:** Total root volume of ‘Ray Ruby’ grapefruit grafted on ‘Kuharske’ citrange planted in flatwood soils located in Fort Pierce, FL, USA, and treated with ground-applied fertilizer, foliar fertilizer and different planting densities from September 2020 to January 2022.

Root Volume (mm^3^)		Sept. 2020	Jan. 2021	May 2021	Sept. 2021	Jan. 2022
Ground-applied ^1^	CRF1	491.8 ± 135	540.8 ± 117	467.1 ± 101	1139.7 ± 232	1949.4 ± 271
	CRF2	732.7 ± 329	439.1 ± 101	316.0 ± 73	1143.3 ± 170	1777.7 ± 285
Foliar ^2^	0×	430.9 ± 199	367.4 ± 131	386.1 ± 138	1044.9 ± 256	1691.2 ± 367
	1.5×	374.4 ± 110	344.8 ± 70	349.2 ± 75	908.5 ± 227	1647.4 ± 316
	3×	856.8 ± 512	607.2 ± 176	285.6 ± 99	1163.3 ± 336	1861.2 ± 443
	6×	692.7 ± 290	755.9 ± 98	672.0 ± 229	1479.3 ± 329	2225.4 ± 452
Planting density ^3^	High	916.5 ± 435	637.0 ± 171	397.5 ± 102	1527.2 ± 312	2131.2 ± 438
	Low	485.4 ± 210	456.4 ± 134	423.4 ± 150	835.7 ± 155	1409.2 ± 161
	Medium	391.6 ± 119	389.4 ± 90	449.9 ± 136	1109.8 ± 249	2072.9 ± 371

^1^ Controlled release fertilizer blends 1 (CRF1): 12N-1.31 P-11.62K and micronutrients at 1× the UF/IFAS recommendation with micronutrients as sulfates (12-3-14 1 × Micro) and CRF2: enhanced 12N-1.31 P-11.62K with 2 × Mg and 2.5× the UF/IFAS recommendation with micronutrients as sulfur-coated products (#12-3-14 2.5 × Micro). ^2^ Foliar fertilizer treatments, which included 0×, 1.5×, 3× and 6× the UF/IFAS recommendation. ^3^ Low (300 trees ha^−1^), medium (440 trees ha−1) and high (975 trees ha^−1^).

**Table 7 plants-12-01659-t007:** *p*-values obtained via a three-way ANOVA followed by Tukey’s post hoc test, corresponding to the total root volume of ‘Ray Ruby’ grapefruit grafted on ‘Kuharske’ citrange planted in flatwood soils located in Fort Pierce, FL, USA, and treated with ground-applied fertilizer, foliar fertilizer and different planting densities from September 2020 to January 2022.

Root Volume *p* Values		Sept. 2020	Jan. 2021	May 2021	Sept. 2021	Jan. 2022
Ground-applied ^1^	CRF1—CRF2	0.82	0.67	0.03	0.98	0.66
Foliar ^2^	1.5×–0×	0.92	0.69	0.65	0.99	0.99
	3×–0×	0.92	0.29	0.88	0.99	0.99
	6×–0×	0.77	0.99	0.97	0.69	0.78
	3×–1.5×	0.99	0.83	0.98	0.92	0.97
	6×–1.5×	0.99	0.62	0.86	0.49	0.72
	6×–3×	0.98	0.26	0.98	0.86	0.92
Planting density ^3^	Low—High	0.22	0.79	0.63	0.09	0.29
	Med—High	0.35	0.97	0.98	0.35	0.97
	Med—Low	0.97	0.69	0.51	0.73	0.38

^1^ Controlled release fertilizer blends 1 (CRF1): 12N-1.31 P-11.62K and micronutrients at 1× the UF/IFAS recommendation with micronutrients as sulfates (12-3-14 1 × Micro) and CRF2: enhanced 12N-1.31 P-11.62K with 2 × Mg and 2.5× the UF/IFAS recommendation with micronutrients as sulfur-coated products (#12-3-14 2.5 × Micro). ^2^ Foliar fertilizer treatments, which included 0×, 1.5×, 3× and 6× the UF/IFAS recommendation. ^3^ Low (300 trees ha^−1^), medium (440 trees ha−1) and high (975 trees ha^−1^).

**Table 8 plants-12-01659-t008:** Total root area of ‘Ray Ruby’ grapefruit grafted on ‘Kuharske’ citrange planted in flatwood soils located in Fort Pierce, FL, USA, and treated with ground-applied fertilizer, foliar fertilizer and different planting densities from September 2020 to January 2022.

Root Area (mm^2^)		Sept. 2020	Jan. 2021	May 2021	Sept. 2021	Jan. 2022
Ground-applied ^1^	CRF1	1598.3 ± 428	1463.1 ± 328	1869.0 ± 420	2763.6 ± 566	4111.4 ± 576
	CRF2	1837.3 ± 498	1452.4 ± 312	1308.2 ± 301	2664.3 ± 402	3773.9 ± 621
Foliar ^2^	0×	1392.9 ± 499	1199.5 ± 457	1580.4 ± 556	2440.6 ± 605	3540.3 ± 771
	1.5×	1240.9 ± 330	1173.8 ± 247	1392.2 ± 300	2126.1 ± 512	3485.3 ± 693
	3×	1770.8 ± 733	1586.1 ± 427	1057.7 ± 354	2733.5 ± 799	3962.2 ± 963
	6×	2388.7 ± 825	2202.7 ± 319	2771.5 ± 973	3633.9 ± 831	4722.7 ± 974
Planting density ^3^	High	2425.8 ± 668	2144.6 ± 532	1644.4 ± 464	3687.8 ± 754	4556.2 ± 949
	Low	1433.9 ± 575	1208.7 ± 252	1686.7 ± 593	1977.3 ± 383	2962.8 ± 363
	Medium	1205.9 ± 260	1039.3 ± 221	1800.6 ± 578	2598.5 ± 587	4364.4 ± 787

^1^ Controlled-release fertilizer blends 1 (CRF1): 12N-1.31 P-11.62K and micronutrients at 1× the UF/IFAS recommendation with micronutrients as sulfates (12-3-14 1× Micro) and CRF2: enhanced 12N-1.31 P-11.62K with 2× Mg and 2.5× the UF/IFAS recommendation with micronutrients as sulfur-coated products (#12-3-14 2.5× Micro). ^2^ Foliar fertilizer treatments, which included 0×, 1.5×, 3× and 6× the UF/IFAS recommendation. ^3^ Low (300 trees ha^−1^), medium (440 trees ha−1) and high (975 trees ha^−1^).

**Table 9 plants-12-01659-t009:** *p*-values obtained via a three-way ANOVA followed by Tukey’s post hoc test, corresponding to the total root area of ‘Ray Ruby’ grapefruit grafted on ‘Kuharske’ citrange planted in flatwood soils located in Fort Pierce, FL, USA, and treated with ground-applied fertilizer, foliar fertilizer and different planting densities from September 2020 to January 2022. Bold values were considered to be significantly different (*p* < 0.05).

Root Area *p* Values		Sept. 2020	Jan. 2021	May 2021	Sept. 2021	Jan. 2022
Ground-applied ^1^	CRF1—CRF2	0.71	0.38	0.03	0.88	0.69
Foliar ^2^	1.5×–0×	0.94	0.37	0.73	0.98	0.99
	3×–0×	0.98	0.16	0.93	0.98	0.98
	6×–0×	0.52	0.99	0.99	0.59	0.77
	3×–1.5×	0.99	0.89	0.98	0.91	0.97
	6×–1.5×	0.86	0.30	0.83	0.40	0.72
	6×–3×	0.67	0.14	0.97	0.79	0.92
Planting density ^3^	Low—High	0.13	0.93	0.81	0.08	0.28
	Med—High	0.36	0.74	0.88	0.28	0.95
	Med—Low	0.85	0.53	0.53	0.76	0.39

^1^ Controlled-release fertilizer blends 1 (CRF1): 12N-1.31 P-11.62K and micronutrients at 1× the UF/IFAS recommendation with micronutrients as sulfates (12-3-14 1× Micro) and CRF2: enhanced 12N-1.31 P-11.62K with 2× Mg and 2.5× the UF/IFAS recommendation with micronutrients as sulfur-coated products (#12-3-14 2.5× Micro). ^2^ Foliar fertilizer treatments, which included 0×, 1.5×, 3× and 6× the UF/IFAS recommendation. ^3^ Low (300 trees ha^−1^), medium (440 trees ha−1) and high (975 trees ha^−1^).

**Table 10 plants-12-01659-t010:** Details about treatment factors and arrangements for experimental design.

Experimental Design	Factor and Level
Main plot	CRF application in the soil 12-3-14 + micronutrients at 1× UF/IFAS recommendation 12-3-14 + 2× Mg + micronutrients (3× UF/IFAS B, Fe, Mn and Zn)
Subplot	Three plant densities Single-row low density (SR/LD): 300 trees per ha Single-row high density (SR/HD): 440 trees per ha Double-row high density (DR/HD): 975 trees per ha
Sub-subplot	Four foliar treatments No supplemental nutrients applied (0×) 1.5 times the recommended doses of B, Mn and Zn ^1^ (1.5×) 3.0 times the recommended doses of B, Mn and Zn ^1^ (3.0×) 6.0 times the recommended doses of B, Mn and Zn ^1^ (6.0×)

^1^ B = 0.28 kg/ha, Zn = 5.6 kg/ha and Mn = 4.2 kg/ha: (UF/IFAS recommendation).

**Table 11 plants-12-01659-t011:** Controlled-release fertilizer formula.

	12-3-14 + Micronutrient at 1× UF/IFAS Recommendation			12-3-14 +2 × Mg + Micronutrients (3× UF/IFAS B, Fe, Mn and Zn)
	Nutrient (%)	Amount (kg ha^−1^)	Nutrient (%)	Amount (kg ha^−1^)
N	12	180	12	180
P_2_O_5_	3	45	3	45
K_2_O	14	209	14	209
Ca	1	15	1	15
Mg	1.2	18	2.4	36
S	13	194	15	228
B	0.1	0.7	0.1	1.7
Cu	0	0.6	0.1	1.5
Fe	0.4	5.9	1	15
Mn	0.6	8.4	1.4	21
Mo	0	0	0	0.2
Zn	0.4	5.9	1	15

## Data Availability

The datasets generated during and/or analyzed during the current study have been deposited to NCBI. The BioProject accession number is PRJNA937287.

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
