# Peer review of "Grapefruit Root and Rhizosphere Responses to Varying Planting Densities, Fertilizer Concentrations and Application Methods"

_plants, 2023, doi:10.3390/plants12081659_

Round 1

Reviewer 1 Report

The manuscript entitled “Foliar and granular fertilization and planting density impacts on root growth and rhizosphere bacterial communities of HLB-affected grapefruit trees'', studies the effects of the impact foliar and granular fertilizer treatments on grapefruit root health and rhizosphere composition at varying planting densities. The authors demonstrated that of use of the CRF2 granular fertilizer resulted in higher soil nutrient concentrations through all time points of the study, notably with significant differences in Zn and Mn. Additionally, increased micronutrient concentrations provided by the CRF2 granular fertilizer may have provided excess nutrients required to assist the root health of HLB-affected grapefruit trees. The article was very well executed and very well written, needing only minor revisions.

Minor suggestions:

The authors need to revise the title of the paper in a more meaningful way.

The introduction does say little about the selected grapefruit plant, why these plants were selected?

Authors should discuss the results integrally. The discussion is based on individual results. I suggest that integrating the results will give more value to the work. I suggest that you discuss by integrating all your results. You can use correlation tests (PCA or Pearson Correlation).

Data Availability Statement: The data sets presented in this study need to be deposited and found in online repositories.

Author Response

The authors need to revise the title of the paper in a more meaningful way.

The title has been changed to better represent the experiment. It now reads: “Grapefruit root and rhizosphere responses to varying micronutrient concentrations and application methods”

The introduction does say little about the selected grapefruit plant, why these plants were selected?

Information has been added to the introduction regarding why grapefruit plants were selected.

“Grapefruit, in particular, are more prone to root dieback when infected with HLB compared to specialty citrus, such as lemon and lime [9]. Most of the grapefruit pro-duction in Florida takes place in the Indian River district, a 200-mile-long area that borders the Atlantic Ocean. Since the introduction of HLB grapefruit production has undergone an 85% decline [10]”.

Authors should discuss the results integrally. The discussion is based on individual results. I suggest that integrating the results will give more value to the work. I suggest that you discuss by integrating all your results. You can use correlation tests (PCA or Pearson Correlation).

Pearson correlation coefficients were calculated and Table 1 (A, B and C) is now added to the result and discussed.

Data Availability Statement: The data sets presented in this study need to be deposited and found in online repositories.

The data has been deposited to NCBI, prior to the submission of the manuscript. The BioProject Accession Number is PRJNA937287. This is now indicated in the paper.

Reviewer 2 Report

Dear Authors of the manuscript on “Foliar and granular fertilization and planting density impacts on root growth and rhizosphere bacterial communities of HLB-affected grapefruit trees” (grafted on Kuharske citrange), and wish to congratulate for the research you are developing since some years.  

It is interesting to know that soil nutrient concentrations were significantly affected by planting density and granular fertilizer treatments.  Changes in root nutrient concentrations and measured parameters were more influenced by granular fertilizer treatments than foliar fertilizer treatments. Wonder if you expscte the same with other rootstocks.

Very interesting are also the results on abundance of Vicinamibacterales and Rhizobiales rhizosphere community, but are not discussed in relation with the nutrient level in the roots or in the soil.   Moreover, since the trees were planted in September 2013, I do not understand if chemical and microbial variation had any effect on HLB disease you cite in the title.

The large amount of data evidence a variability of single macro- and micro-nutrients concentration, which, surprisingly, vary up and down at different samplings. Root concentration of Mn and Zn was greater at high density than medium density just in May 2021 and January 2022, not in other times. Similar profiles are reported for other nutrients. I wonder if a monitoring longer than two years would confirm such variations and we must consider them as preliminary results.

I'm also doubtful if some complex graphics add useful information, or rather if in some cases they are not easily understandable. Please, consider if focusin them only on nutrients significatively different would make the reading simpler and more attractive.

In conclusion, I agree with you that “further insight into the interactions shared between fertilizer regimens, rhizosphere microbial composition, and root health is still needed, specific to citrus”, and  would suggest to make minor changes, in order to let catch more easily the value of the results.

Author Response

Dear Authors of the manuscript on “Foliar and granular fertilization and planting density impacts on root growth and rhizosphere bacterial communities of HLB-affected grapefruit trees” (grafted on Kuharske citrange), and wish to congratulate for the research you are developing since some years.  

Thanks so much for your kind words.

It is interesting to know that soil nutrient concentrations were significantly affected by planting density and granular fertilizer treatments.  Changes in root nutrient concentrations and measured parameters were more influenced by granular fertilizer treatments than foliar fertilizer treatments. Wonder if you expscte the same with other rootstocks.

There have been studies on sweet oranges grafted on different rootstocks and grown in different Florida regions that reported a more pronounced change in micronutrient uptake and translocation. Additionally, there are some studies on novel scion/rootstock combinations in our areas that are showing promising results. Overall, grapefruit trees are the most sensitive to the citrus greening (HLB) disease, and this is probably the reason of the lower impact of micronutrient fertilization on the tree uptake and response.

Here are some related publications:

  • https://doi.org/10.3390/plants12010073
  • https://doi.org/10.3390/plants11233226
  • https://doi.org/10.3390/horticulturae8090763
  • https://doi.org/10.1002/saj2.20497
  • https://doi.org/10.3390/horticulturae8111027

Very interesting are also the results on abundance of Vicinamibacterales and Rhizobiales rhizosphere community, but are not discussed in relation with the nutrient level in the roots or in the soil.   Moreover, since the trees were planted in September 2013, I do not understand if chemical and microbial variation had any effect on HLB disease you cite in the title.

Thanks for your suggestions. The paper mainly focuses on compositional differences of rhizosphere bacteria, which relate to alpha and beta diversity parameters rather than abundances. The study would have needed to be better suited to analyzing these differences in rhizosphere bacterial abundance related to fertilizer treatments. but we are in agreement with your comment and the word HLB has been removed from the title as suggested. So, the paper now has a more accurate title.

The large amount of data evidence a variability of single macro- and micro-nutrients concentration, which, surprisingly, vary up and down at different samplings. Root concentration of Mn and Zn was greater at high density than medium density just in May 2021 and January 2022, not in other times. Similar profiles are reported for other nutrients. I wonder if a monitoring longer than two years would confirm such variations and we must consider them as preliminary results.

Due to citrus being a perennial tree crop, the season of the sampling periods can have an affect on the concentrations of nutrients in the roots and soil. The trees were planted in September 2013, and since other studies have already been conducted on this experimental plot ( see citations 9 and 22) we can not really consider our data as preliminary. Another variable affecting these results is the progression of HLB disease, which may cause variation as well over time.

I'm also doubtful if some complex graphics add useful information, or rather if in some cases they are not easily understandable. Please, consider if focusin them only on nutrients significatively different would make the reading simpler and more attractive.

We noticed much of this issue was caused by nutrients, such as Ca, not scaling properly with other nutrients on figures. Changes have been made to better address this issue and Pearson correlation coefficients have been calculated and added to the manuscript.

In conclusion, I agree with you that “further insight into the interactions shared between fertilizer regimens, rhizosphere microbial composition, and root health is still needed, specific to citrus”, and  would suggest to make minor changes, in order to let catch more easily the value of the results.

We valued your comments and your suggested changes have been made to the result section and to the paper.

Reviewer 3 Report

Dear authors,

The topic is interesting, so it was interesting to read the article.

It was difficult to understand some results. To understand the abstract it is necessary to read the body of the article, which should not happen.

Graphs and tables should be improved. The graphs show mineral elements that should be presented with different scales.

In data analysis, the split-split-plot design of the test is not taken advantage of. It remains to make a correlation between the applied nutrients and the concentrations of these nutrients in the soil and in the roots. The way of presenting the results should therefore be much clearer.

It appears that part of the trial results were presented in another article (reference 21). This makes the article less interesting, because we lack the perspective of what happened in the aerial part of the plant.

Perhaps it would be important to relate the root part data with the results in the aerial part of the plant and production. For this, it is not necessary to repeat the data presented in the other article, although the repetition of some values in this case would be justified.

Nor is it clear why the results of the same trial have so few authors in common between this article and reference 21.

More detailed comments were made in the article file. Some are mostly suggestions for improving the article.

Author Response

It was difficult to understand some results. To understand the abstract it is necessary to read the body of the article, which should not happen.

The abstract has been rewritten to address this issue.

Graphs and tables should be improved. The graphs show mineral elements that should be presented with different scales.

The graphs have been improved and now calcium has its own graphs. Additionally, boron was also separated in the root micronutrient graphs to better adjust for differences in scales.

In data analysis, the split-split-plot design of the test is not taken advantage of. It remains to make a correlation between the applied nutrients and the concentrations of these nutrients in the soil and in the roots. The way of presenting the results should therefore be much clearer.

Pearson correlation coefficients were calculated, and Table 1 (A, B and C) is now added to the result and discussed.

It appears that part of the trial results were presented in another article (reference 21). This makes the article less interesting, because we lack the perspective of what happened in the aerial part of the plant.

Both trials were conducted at separate times using the same experimental plots. This study focused on the belowground components from 2019-2022, whereas the aboveground component was analyzed by a separate research group from 2015-2019.

Perhaps it would be important to relate the root part data with the results in the aerial part of the plant and production. For this, it is not necessary to repeat the data presented in the other article, although the repetition of some values in this case would be justified.

Since both the aerial and belowground data were taken at different timepoints, correlations likely wouldn’t be representative.

Nor is it clear why the results of the same trial have so few authors in common between this article and reference 21.

Both studies have been conducted on the same experimental field at the UF/IFAS Indian River Research and Education Center, in Fort Pierce, FL, but at separate times and by two different graduate students working in two different labs. Dr. Ferrarezi and Dr. Kadyampakeni are the two authors in common with both publications because they were the lead PIs of the field experiments.

More detailed comments were made in the article file. Some are mostly suggestions for improving the article.

All the comments in the article file have been addressed.

Round 2

Reviewer 3 Report

Dear authors,

The title and abstract were improved. The description of the trials became clearer. However, the presentation of the results remains inadequate and difficult to analyze.

The presentation of nitrogen and phosphorus contents side by side (in figure 5, for example) is not understandable when they are not comparable and this causes the phosphorus bars to be so small that it is impossible to see the differences between treatments.

Differences between treatments that are not consistent are described. In the discussion, these differences are considered to be due to treatments, when, in my view, they are mostly random. It is not understood, and no explanation is given, why a nutrient is higher at one date in one treatment and higher at another date in another treatment.

Comments by line:

Line 16: "caused severe decline"

Line17: Manage the health or the productivity/performance?

Line 21: Citrus trifoliata

Line 38: huanglongbing - Do not capitalize disease names

Line 98: The differences between treatments are not consistent; differ on one sampling date but not on others. Furthermore, it seems to me difficult to explain these differences. The opening sentence seems too assertive to me.

Graphics: Graphics remain very difficult to read/interpret. When we want to see how the content of a nutrient or other parameter evolved, we have to look for the bars, seeing the legend at the bottom, leaving many bars in the middle that are not relevant for this comparison.

Author Response

Dear authors,

The title and abstract were improved. The description of the trials became clearer. However, the presentation of the results remains inadequate and difficult to analyze.

The presentation of nitrogen and phosphorus contents side by side (in figure 5, for example) is not understandable when they are not comparable, and this causes the phosphorus bars to be so small that it is impossible to see the differences between treatments.

Thanks for your time. We addressed all your comments below.

Differences between treatments that are not consistent are described. In the discussion, these differences are considered to be due to treatments, when, in my view, they are mostly random. It is not understood, and no explanation is given, why a nutrient is higher at one date in one treatment and higher at another date in another treatment.

A sentence was added to the discussion about the possible reason why a lack of consistency was observed during the time periods. It can be found on Line 418.

“The phenology of citrus trees with several vegetive flushes during the year, in combination with the sub-tropical weather of Florida (with abundance of rain, during the summer and fall seasons) and the deterioration of tree health caused by HLB disease, may have contributed towards the inconsistency in nutrients concentrations, and the lack of observed patterns during the time periods of the study.”

Comments by line:

Line 16: "caused severe decline"

Fixed.

Line17: Manage the health or the productivity/performance?

Fixed.

Line 21: Citrus trifoliata

Fixed.

Line 38: huanglongbing - Do not capitalize disease names

Fixed.

Line 98: The differences between treatments are not consistent; differ on one sampling date but not on others. Furthermore, it seems to me difficult to explain these differences. The opening sentence seems too assertive to me.

Changes were made throughout the result section to clarify that there are significant differences, but no clear patterns were established in between seasons.

Graphics: Graphics remain very difficult to read/interpret. When we want to see how the content of a nutrient or other parameter evolved, we have to look for the bars, seeing the legend at the bottom, leaving many bars in the middle that are not relevant for this comparison.

Figure 5 was split into two new figures, so now the graphs are clearer to understand. With the amount of information and the different timepoints it is difficult to report the data in a better way. We are confident that the newly added graphs help alleviate the challenges in interpreting the graphs.